# Sox10-Deficient Drug-Resistant Melanoma Cells Are Refractory to Oncolytic RNA Viruses

**DOI:** 10.3390/cells13010073

**Published:** 2023-12-29

**Authors:** John Abou-Hamad, Jonathan J. Hodgins, Edward Yakubovich, Barbara C. Vanderhyden, Michele Ardolino, Luc A. Sabourin

**Affiliations:** 1Centre for Cancer Therapeutics, Ottawa Hospital Research Institute, 501 Smyth Road, Ottawa, ON K1H 8L6, Canada; joabou@ohri.ca (J.A.-H.); eyakubovich@toh.ca (E.Y.); bvanderhyden@ohri.ca (B.C.V.); mardolino@ohri.ca (M.A.); 2Department of Cellular and Molecular Medicine, University of Ottawa, Ottawa, ON K1H 8L6, Canada; 3Department of Biochemistry, Microbiology and Immunology, University of Ottawa, Ottawa, ON K1H 8L6, Canada

**Keywords:** melanoma, SOX10, BRAFV600E, oncolytics, vemurafenib, drug resistance

## Abstract

Targeted therapy resistance frequently develops in melanoma due to intratumor heterogeneity and epigenetic reprogramming. This also typically induces cross-resistance to immunotherapies. Whether this includes additional modes of therapy has not been fully assessed. We show that co-treatments of MAPKi with VSV-based oncolytics do not function in a synergistic fashion; rather, the MAPKis block infection. Melanoma resistance to vemurafenib further perturbs the cells’ ability to be infected by oncolytic viruses. Resistance to vemurafenib can be induced by the loss of SOX10, a common proliferative marker in melanoma. The loss of SOX10 promotes a cross-resistant state by further inhibiting viral infection and replication. Analysis of RNA-seq datasets revealed an upregulation of interferon-stimulated genes (ISGs) in SOX10 knockout populations and targeted therapy-resistant cells. Interestingly, the induction of ISGs appears to be independent of type I IFN production. Overall, our data suggest that the pathway mediating oncolytic resistance is due to the loss of SOX10 during acquired drug resistance in melanoma.

## 1. Introduction

Targeted therapies revolutionized melanoma treatment due to their superior efficacy and specificity. Vemurafenib, an ATP analog, was the first direct BRAF^V600E^ targeted therapy that showed great specificity and increased overall survival of melanoma patients [1,2,3,4]. Unfortunately, the effectiveness of these therapies lasts only for about a year, after which most melanoma patients begin to relapse [5,6]. Other MAPK inhibitors (MAPKi) have been used in conjunction with BRAF inhibitors, but resistance still occurs [7,8]. This is in part due to the large intratumor heterogeneity of melanoma, where some cells can undergo epigenetic alterations and shift their gene expression profile during prolonged treatment [9,10].

Interestingly, targeted therapy resistance can be induced through the loss of SOX10, a key player in melanoma initiation and progression [11,12,13,14,15]. Studies have also shown that the loss of SOX10 increases the cancer stem cell properties of melanoma, which could be the underlying factor in creating this resistant state [16,17]. As more work unravels this concept of cancer stem cells, others have observed that the enrichment of cancer stem cells induces a cross-resistant state to both chemotherapy and immunotherapy [18]. Whether targeted therapy induces a cross-resistance environment to all forms of therapy remains to be assessed. 

A more recent method of melanoma treatment is an oncolytic virus-derived therapeutic [19]. One virus, namely vesicular stomatitis virus (VSV), has been well documented to target and kill proliferative cancers, such as melanoma, both in vitro and in vivo [20,21,22]. In healthy cells, viral infection induces a type I interferon (IFN) response, which increases the production of IFNα/β [23]. This in turn activates specific interferon-stimulated genes (ISGs) that are used to repress viral replication and spread [23,24]. This response is turned off in cancer cells since the IFN pathway inhibits growth, allowing oncolytic viruses such as VSV to specifically target cancer cells [25]. Although VSV treatment has been shown to readily kill melanoma cells, we tested whether targeted therapy could be used together with oncolytic viruses.

In this study, we find that VSV and MAPKis do not function synergistically; rather, rather MAPKis block VSV infection. We further show that melanoma cells that acquire a drug-resistant state are less susceptible to viral infection, and this cross-resistant state can be recapitulated following the deletion of SOX10. Finally, we find that both the induced resistant state and the loss of SOX10 induce the activation of ISGs independent of IFNα/β, priming the cells for resistance to infection by oncolytic viruses. These data suggest that a therapeutic regimen that supports a drug-sensitive melanocytic state could be enhanced by the addition of oncolytics.

## 2. Materials and Methods

### 2.1. Cell Culture

The Yale University Mouse Melanoma cell line YUMM1.1 and the human melanoma cell line A375 were a kind gift from Dr. William Damsky and Dr. John Copeland (respectively). Vero cells were a kind gift from Dr. Michele Ardolino, and the 293Ts were obtained from ATCC. All cell lines were cultured in DMEM supplemented with 10% FBS, 1% L-glutamine, and 1% penicillin/streptomycin. The YUMM1.1 cell line also received a supplement of 1X non-essential amino acids.

### 2.2. Establishment and Isolation of Vemurafenib-Resistant Cell Populations

Melanoma cell lines were plated in 60 mm dishes at 40% confluency and allowed to adhere overnight. The next day, cells were treated with vemurafenib at their respective IC50 for 6 weeks, with fresh media and drug replacement occurring every three days. The cells were then taken off the drug treatment and allowed to recover and retested for vemurafenib sensitivity for three days to confirm the resistant phenotype.

### 2.3. Oncolytic Infection

The melanoma cells were plated in multiple wells, left to adhere overnight, and counted the next day from duplicate wells. The cells were then infected with various viruses at various MOIs (MOI = $\frac{\mathrm{Number}\;\mathrm{of}\;\mathrm{cells}}{\mathrm{amount}\;\mathrm{of}\;\mathrm{liquid}\;\mathrm{added}\;\mathrm{per}\;\mathrm{well}\;(L)}*\mathrm{desired}\;\mathrm{MOI}*\mathrm{amount}\;\mathrm{required}$). Viral supernatant was added to each well for 1 h at 37 °C, removed, and replaced with full media for 24 h. The cells were then collected and quantified for YFP or eGFP by flow cytometry.

### 2.4. Viral Titer

Viral supernatant from infected cells was collected 24 h post infection and overlaid onto confluent Vero cells in multiple serial dilutions and incubated at 37 °C for 90 min. To allow for virus adherence, a 1:1 mix of 1% agar and 2X DMEM with 20% FBS was added to each well and allowed to solidify at room temperature for 10 min. Plates were incubated at 37 °C for 24 h and fixed (3:1 methanol:glacial acetic acid) for one hour. Agar overlays were washed away using water, the wells were stained with Coomassie blue, and plaque forming units were counted. 

### 2.5. Resazurin Viability Assay

The melanoma cells were plated in a 96-well plate and treated with various compounds for 48 (any viral infection) or 72 h (vemurafenib, dabrafenib, and/or trametinib; SelleckChem, Houston, TX, USA). Control wells supplemented with PBS and/or DMSO (viral infection or MAPKi, respectively) were used as a baseline for 100% viability. At the end of the treatment, the supernatants in each well were removed and replaced with 100 µL of resazurin solution (55 µM resazurin salt, 10% FBS, 1X DMEM). The absorbance of each well was read at 570 and 604 after four hours.

### 2.6. SiRNA Transfection and Lentivirus Production

All siRNAs used were purchased from Dharmacon. Lipofectamine 3000 (Invitrogen, Waltham, MA, USA) was used to introduce 200 nM of siRNA into the YUMM1.1 cells. Transfected YUMM1.1 cells were seeded into a 96-well dish 24 h post transfection and subsequently treated with vemurafenib at various concentrations 24 h post plating.

Viral particles were produced by seeding 293T cells (5 × 10^6^) in a 10 cm dish. 293Ts were transfected using lipofectamine 3000 with 10 µg of construct, 8 µg of pCMV-dR8.2 dvpr, and 2 µg of pCMV-VSV-G. The next day, the medium was changed and viral supernatant was collected 72 h later and passed through a 0.44 µm filter.

### 2.7. Lentiviral-Mediated CRISPR/Cas9 Knockouts

The melanoma cells were infected for 6 h with 1 mL of viral supernatant diluted in 5 mL of cell culture media supplemented with 10 µg/mL of polybrene. The medium was replaced with full medium, and the cells were selected with 5 µg/mL of blasticidin (for Cas9 plasmid) or sorted for RFP (for CRISPR guides) 48 h post infection. Knockout cells were confirmed using Western blotting. Sequences for mouse sgSOX10 were as previously described [16]. Human guide RNAS have the following sequences: sgSOX10-4: GATGGAGCGCCCGT-CCCGCT and sgSOX10-24: GTGACAAGCGCCCCTTCATC.

### 2.8. Cell Lysis and Western Blotting

The melanoma cell lines were lysed using RIPA lysis buffer containing protease and phosphatase inhibitors [16]. Lysates were vortexed at increments of 5 min on ice for a total of 30 min or freeze-thawed to allow for complete isolation of whole cell protein. Lysates were cleared at 16,000× *g* for 10 min at 4 °C. Bradford reagent (BioRad) was used to quantify protein concentration. Equivalent amounts of protein were denatured, electrophoresed on polyacrylamide gels, and transferred onto PVDF (polyvinylidene difluoride) membranes. The membranes were probed with primary antibodies in 5% bovine serum albumin (BSA) in 1X TBST overnight at 4 °C, followed by HRP-conjugated secondary antibodies. Reactive bands were detected with enhanced chemiluminescence reagent and exposure to X-ray films. The membranes were probed with SOX10 (CellSignaling Technology, Danvers, MA, USA; Cat:89356), custom pan-VSV (a kind gift from Dr. Jean Simon Diallo), and β-actin (Sigma, Oakville, ON, Canada; Cat:A5316). 

### 2.9. RNA-Sequencing, EnrichR Biological Pathway Analysis and TPM Analysis

Data from the A375 control and SOX10-KO-2 were collected from NCBI BioProject PRJNA748713. Transcript quantification was executed using Kallisto (v0.45.0) [26] with the GRCh38 build of the human transcriptome and the -b 50 bootstrap option. Raw counts were imported into DESeq 2 (1.40.2) and analyzed [27]. Genes with less than 10 counts were removed. Differential gene expression with an adjusted *p*-value < 0.05 was identified and fold change was determined comparing the A375 sgNTC samples with the SOX10 KO-2 samples.

Genes upregulated in the SOX10 knockout cells were imported into EnrichR to determine the biological processes activated within the SOX10 knockout cells [28,29]. Only genes that had a *p*-value < 0.05 were used during this analysis. The top 14 most significant biological processes (smallest *p*-value = largest−log_10_(*p*-value) are shown in Figure 4B. Then, the 14 most significant viral (defined by any process including the “type I IFN” or “viral” in its title) processes (which were taken from the full list of biological processes) were used to create Figure 4C. Genes found within the gene sets are from MSigDb [30].

To determine whether the correlation observed with the SOX10 KO and the ISGs (Table 1) also occurred in targeted therapy-resistant cells, we used deposited data from NCBI BioProject PRJNA748714. TPM values for each gene were used, and relative expression was calculated using a Z-score. 

### 2.10. Statistical Analysis

All graphs and statistical analysis were done using GraphPad Prism. Data are represented in mean ± SEM and significance was calculated using a two-tailed *t*-test, where * *p* < 0.05, ** *p* < 0.01, and *** *p* < 0.001.

## 3. Results

### 3.1. MAPKi Treatment Blocks VSV Infection in BRAF^V600E^ Melanoma

To determine whether VSV oncolytic therapy could be used in conjunction with current therapies, we tested the combination of VSVΔ51 [31] with MAPK inhibitors. We treated the Sox10-expressing murine YUMM1.1 cells (which harbor a BRAF^V600E^ mutation) with VSVΔ51 and with either vemurafenib (5 µM), dabrafenib (1 µM), or trametinib (0.1 µM) (Figure 1A). Interestingly, we observed a reduction in infection (measured using a viral YFP reporter) when the cells were co-treated with BRAF and MEK inhibitors (Figure 1A). When tested in the A375 human melanoma cell line, we observed a similar reduction in viral infection with BRAF/MEK inhibitors (Figure 1A). To confirm these findings, we measured viral replication using plaque assays. In agreement with our viral YFP reporter infection data, we observed an approximately 100-fold decrease in viral titers when YUMM1.1 and A375 melanoma cells were co-treated with MAPKi relative to VSVΔ51 alone (Figure 1B). As expected, there was a significant decrease in viability in cells infected with VSVΔ51 only and cells treated with BRAF/MEK inhibitors only (Figure 1C). However, viability was not significantly altered by co-treatment with BRAF/MEK inhibitors and VSVΔ51 relative to the individual treatments, suggesting that MAPK inhibitors block VSV infection and lytic function (Figure 1C). Therefore, MAPK-targeted therapies in melanoma block the in vitro efficacy of the oncolytic VSVΔ51.

### 3.2. Vemurafenib-Induced Targeted Therapy Resistance Promotes a Cross-Resistant State between Other MAPKis and VSV Infection

Resistance to targeted therapy is common in clinical settings [5,6]. Given that concurrent treatment with MAPKi blocks VSV infection (Figure 1), we determined whether MAPK-resistant melanomas are also resistant to viral infection/lysis. To test this, we generated three independent vemurafenib-resistant populations by chronically treating YUMM1.1 and A375 cell lines for six weeks at their IC_50_ concentrations (5 µM and 2 µM, respectively; Figure 2A). Three independent populations were generated with varying resistance to vemurafenib but also to other MAPKi inhibitors (Figure 2B–D and Appendix A). Drug-resistant cells were then tested for VSVΔ51 infection. Interestingly, we found that all drug-resistant populations had also acquired resistance to viral infection, with the most dramatic phenotype observed in the YUMM1.1 ResA cells (Figure 2E). This was also observed in human A375 melanoma cells, although not as robust as A375 drug-resistant cells, which have a tendency to revert back to wild-type tolerance following brief culture in the absence of vemurafenib treatment (Appendix A). This suggests a tight link between MAPKi resistance and resistance to viral infection. Plaque assays showed reduced viral titers (Figure 2F) and resistance to virus-induced cell killing (Figure 2G), supporting the notion that MAPKi resistance in melanoma further induces viral therapy resistance against VSVΔ51. This resistance was also assessed in YUMM1.1 cells treated acutely for 3 or 9 days with vemurafenib. After rechallenge with vemurafenib or VSVΔ51 we observed no major differences in VSVΔ51 infection or viability, suggesting that long-term transition or reprogramming is required to promote this cross-resistant state (Appendix A). Therefore, resistance to MAPK-targeted therapies in melanoma also promotes cross-resistance to oncolytic virus therapy.

### 3.3. SOX10 Is Lost during Chronic Targeted Therapy Treatment and Induces a Cross-Resistant State to VSVΔ51

Given the prevalence of resistance to MAPK-targeted therapies, several genes and pathways potentially implicated in this process have been identified [32,33]. We reasoned that these same resistance mechanisms may also play a role in cross-resistance to oncolytic virus therapy. We have previously shown that the transcription factor SOX10, a major marker of primary melanomas and the melanocytic state, reduces the cancer stem cell properties of melanoma cells [16]. Others have found that the loss of SOX10 expression induces MAPKi resistance [11,12,13,14,15,16]. Interestingly, SOX10 has been associated with the regulation of immune-related pathways, including IRF1, a well-characterized regulator of viral infection [34]. Therefore, we hypothesized that the loss of SOX10 in MAPKi-resistant cells could induce resistance to oncolytic virus infection. Chronic vemurafenib treatment led to the loss of SOX10 in melanoma cells and a marked reduction in VSVΔ51 infection (Figure 3A). We then tested VSVΔ51 infection in SOX10-depleted cells. Using siRNAs, an 80–90% SOX10 knockdown was consistently observed and promoted resistance to vemurafenib treatment (Figure 3B,C). We further tested VSV infection and replication in previously reported stable YUMM1.1 SOX10-null cells [16] and A375 SOX10 knockout cells. In both SOX10-deficient cells, a substantial reduction in VSVΔ51 infection and titers was observed, suggesting that SOX10 plays a key role in resistance to oncolytic virus therapy (Figure 3D,E). In contrast, decreased virus-induced cell death was seen in SOX10 knockout cells (Figure 3F). Therefore, SOX10 expression is required for efficient VSVΔ51 infection in melanoma cells, and its expression is downregulated following resistance to MAPK inhibitors. 

The cellular response to RNA viruses, such as VSVΔ51 is notably different from the response to DNA viruses. We then asked if the cross-resistance to oncolytic virus therapy after MAPKi resistance is unique to RNA viruses. To this end, we expanded our panel of viruses to include another oncolytic RNA virus, Maraba MG1, and the DNA virus vaccinia (VVTT). As expected, our drug-resistant melanoma cells and SOX10 knockout cells were resistant to MG1 but not to vaccinia virus infection (Appendix A), suggesting that this phenotype is conserved among RNA viruses and that these cells become “primed” to RNA virus infection upon acquiring a drug-resistant state.

Although many mechanisms of resistance to viral infection exist, the most prominent mechanism arises from the type I interferon pathway [35]. Since VSVΔ51 harbors a mutation that sensitizes it to the anti-viral effects of type I IFNs, we examined the role of that pathway in cross-resistance to oncolytic virus therapy. First, we made use of wild-type VSV, which antagonizes the production of type I IFNs in infected cells. However, wild-type VSV infection was similarly reduced in drug-resistant melanoma cells and SOX10 KO cells, analogous to our data with VSVΔ51 (Appendix A). Next, we treated the cells with the JAK1/2 inhibitor ruxolitinib (RUXO), a blocker of the type I IFN response. Treatment with RUXO restored VSVΔ51-mediated killing and replication to control levels in most drug-resistant cells and the SOX10 KO cells (Appendix A). Interestingly, the sensitivity of the highly resistant YUMM1.1 ResA population was restored to wild-type levels following co-treatment with VSV and RUXO (Appendix A). This suggests that the drug-resistant state acquired through chronic treatment or via the loss of SOX10 alters the type I IFN pathway to block oncolytic RNA viruses.

### 3.4. RNA-Sequencing Analysis of A375 SOX10 Knockout Cells Shows Enrichment of Viral Responses Gene Sets

To gain further insight into the resistant mechanism to VSV infection, we re-analyzed RNA-sequencing data from SOX10 knockout A375 cells [15]. This dataset shows that there are approximately 3600 upregulated genes in the SOX10 knockouts and 3300 downregulated genes (Figure 4A). By using EnrichR, we identified various biological processes that are acquired within the SOX10 knockout populations [28,29]. Of the top 14 enriched biological processes in the SOX10-deficient A375, two were associated with the negative regulation of viral infection (Figure 4B). We further mined any significant biological processes that had any impact on viral susceptibility and propagation and identified 14 of them (Figure 4C).

We next evaluated four of these viral gene sets (Figure 4C) (defense response to virus, negative regulation of viral process, negative regulation of viral genome, and regulation of viral entry into the host cell) to determine which genes are differentially expressed between SOX10 knockouts and controls. In all four viral gene sets, we found more viral genes to be upregulated in the knockouts compared to controls (Figure 4D,G). We compared all viral genes from our four gene sets that were upregulated in the SOX10 knockout cells and organized them into a Venn diagram to determine gene overlap between the gene sets (Figure 4H). Interestingly, IFITM1/2/3, and TRIM6, common to all four gene sets, were shown to repress viral infection (Table 1) [36,37]. Many of the genes identified within the viral biological processes were ISGs, which become activated post viral infection (Table 1). Interestingly, we do not observe IFNα/β within our list, suggesting that alternative pathways activate the ISGs (Table 1). We also corroborated the activation of those genes in targeted therapy-resistant PBRT cells where anti-viral genes are also induced (Table 1). This was also corelated with SOX10 downregulation (Appendix A). Together, our data suggest that the loss of SOX10 during targeted therapy treatment induces the activation of ISG genes, allowing for cross-resistance to RNA-based oncolytic viruses, in addition to MAPKis.

## 4. Discussion and Conclusions

We find that acquired resistance to vemurafenib further induces a cross-resistant state to VSV oncolytic treatment (Figure 2). This is also observed within our SOX10 knockout cells (Figure 3). Interestingly, we find a large variation in VSV resistance within our YUMM1.1 resistant lines (ResA-C), where we observe minimal infection at MOI 10 within our YUMM1.1 ResA cell line (Figure 2E). Surprisingly, this lack of infection could not be enhanced after using ruxolitinib, which restored ResB and ResC infection back to wild-type levels (Appendix A). Although no increase in infection was observed following a 24 h infection, cell viability measurements 48 h post infection restored cell sensitivity to wild-type levels during co-treatments (Appendix A). Although this finding was unexpected, it does suggest that the ResA’s may express a high baseline level of ISGs that may require prolonged treatment with Ruxo to completely turn off the JAK-STAT pathway and induce viral sensitivity.

Our data suggest that the transition from a wild-type to a targeted therapy-resistant state induces an IFN-like response where the cell upregulates the expression of many ISGs that is independent of IFNα/β induction (Figure 4, Table 1, Appendix A). Surprisingly, this environment can be recapitulated through the loss of SOX10, suggesting that the loss of SOX10 results in the activation of these ISGs (Figure 4). Interestingly, the loss of SOX10 also enriches the cancer stem-like state [16] and induces a cross-resistant melanoma state, no longer being responsive to both targeted therapies and immunotherapies [17,18]. Consistent with these phenotypes, this resistant state can be extended to RNA-based oncolytics.

Recent findings from our lab and others suggest that SOX10 acts as a gatekeeper to the undifferentiated state [12,15,38]. One possibility is that SOX10 does not directly inhibit these genes but rather acts indirectly by blocking the cells’ transition to an undifferentiated state. This undifferentiated state is key to cross-resistance and has been identified as a TEAD+/AP-1+/SOX9+ state [12,38]. The heterogeneity and complex reprogramming observed in this state make it refractory to current therapies. The identification of novel therapeutics that would maintain an SOX10+ state would likely be beneficial. These could be used in combination with existing therapies by preventing reprogramming and the loss of SOX10 expression and the differentiated state.

## Figures and Tables

**Figure 1 cells-13-00073-f001:**
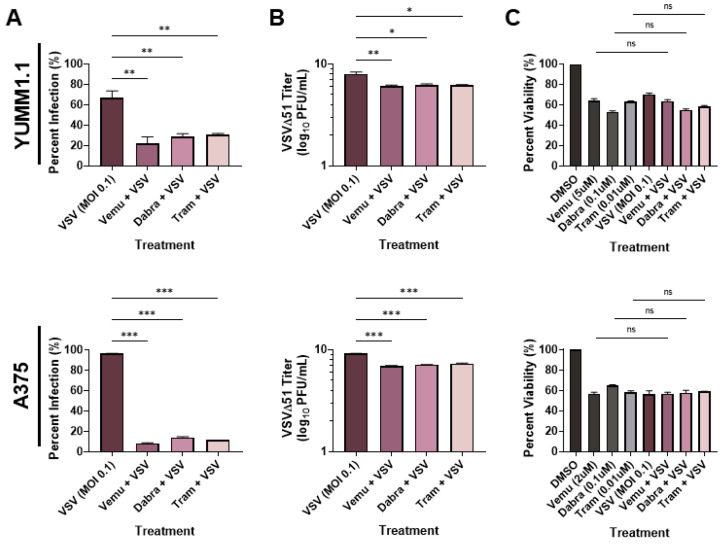
MAPKi treatment blocks VSV infection in BRAF^V600E^ melanoma. YUMM1.1 and A375 cell lines were co-treated with VSVΔ51 at an MOI of 0.1 and vemurafenib (5 µM or 2 µM respectively), dabrafenib (0.1 µM), or trametinib (0.01 µM) for (**A**,**B**) 24 or (**C**) 48 h. (**A**) Quantification of YFP-positive melanoma cells 24 h post MAPKi and/or VSVΔ51 treatment. (**B**) Quantification of viral titers by plaque assays 24 h post infection. (**C**) Cell viability assay (AlamarBlue) 48 h post single and co-treatments. (**A**–**C**) All data are represented as a mean ± SEM of biological triplicates. (**A**,**B**) Significance was calculated using a two-tailed *t*-test, where * *p* < 0.05, ** *p* < 0.01, and *** *p* < 0.001. (**C**) Significance was calculated using a one-way ANOVA. ns, not significant.

**Figure 2 cells-13-00073-f002:**
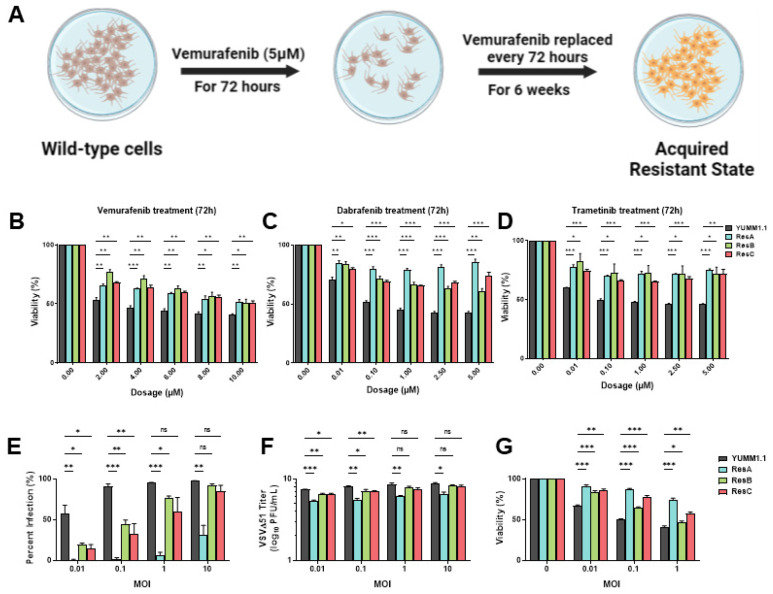
Vemurafenib-induced targeted therapy resistance promotes a cross-resistant state between other MAPKis and VSV infection. (**A**) Schematic representing the 6 week treatment regimen to convert the parental YUMM1.1 cell line into a vemurafenib-resistant cell state. (**B**–**D**) YUMM1.1 and resistant cell lines were seeded in at least technical triplicates and treated at an increasing concentration of vemurafenib (**B**), dabrafenib (**C**), or trametinib (**D**). Cell viability was assayed using AlamarBlue 72 h post treatment. (**E**–**G**) Quantification of (**E**) YFP-positive cells (**E**), viral titers (**F**), and cell viability (**G**) of YUMM1.1 and Res cells 24 (**E**,**F**) or 48 h (**G**) post VSVΔ51 infection. All data are represented as a mean ± SEM of biological triplicates. Significance was calculated using a two-tailed *t*-test, where * *p* < 0.05, ** *p* < 0.01, and *** *p* < 0.001. ns, not significant.

**Figure 3 cells-13-00073-f003:**
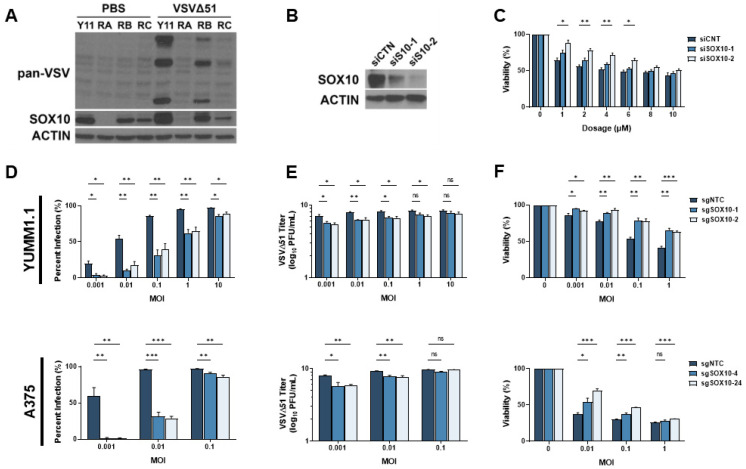
SOX10 is lost during chronic targeted therapy treatment and induces a cross-resistant state to VSVΔ51. (**A**) Immunoblot of YUMM1.1 (Y11) and vemurafenib-resistant cells ResA (RA), ResB (RB), and ResC (RC) 24 h post infection with VSVΔ51. (**B**) Immunoblot of YUMM1.1 transfected with siRNA targeting a random sequence (siCTN) or Sox10 (siS10-1 or siS10-2). (**C**) YUMM1.1 cells transfected with siRNAs targeting Sox10 were treated with various concentrations of vemurafenib, and viability assays were performed 72 h post treatment. (**D**–**F**) YUMM1.1 and A375 cell lines were infected with Cas9 and sgRNA targeting SOX10 to create 2 independent SOX10 knockout populations per cell line. (**D**) Control and SOX10 knockout cells were infected with VSVΔ51 for 24 h, and YFP-positive cells were quantified. (**E**) Viral titers were quantified by counting plaques 24 h post infection. (**F**) Control and SOX10 knockout cells were infected with VSVΔ51 and cell viability was quantified 48 h post infection with AlamarBlue. All data are represented as a mean ± SEM of biological triplicates. Significance was calculated using a two-tailed *t*-test, where * *p* < 0.05, ** *p* < 0.01, and *** *p* < 0.001. ns, not significant.

**Figure 4 cells-13-00073-f004:**
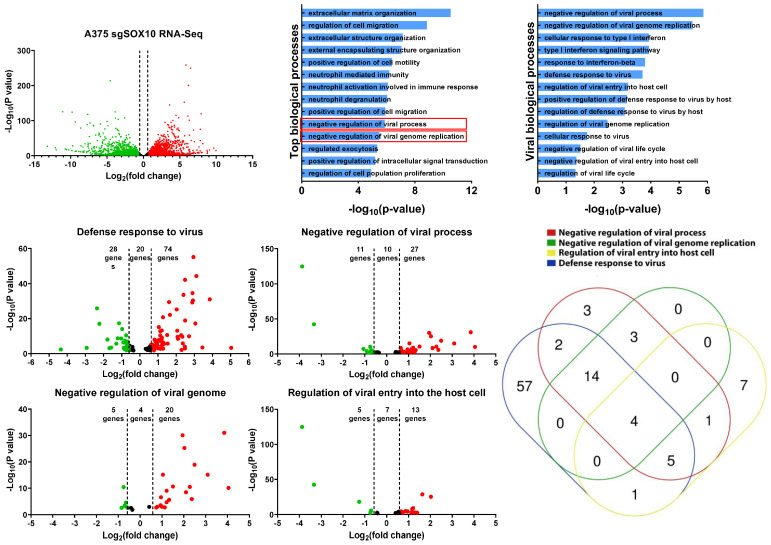
RNA-sequencing analysis of A375 SOX10 knockout cells shows enrichment of viral response gene sets. (**A**) The volcano plot illustrating 6900 significantly differentially regulated genes, where in the red dots represent upregulated in the A375 SOX10 knockout cells and the green dots represent downregulated the wild-type. The black represents the genes that lack an upregulation of more than 1.5-fold. (**B**) GO enrichment analysis of biological processes from the genes upregulated in the SOX10 knockout populations from. Viral processes are in red boxes. (**A**). (**C**) Arrangement of all viral biological processes observed within the A375 Sox10 knockout cells (only viral biological processes –log_10_(*p*-value) > 1 is shown). (**D**–**G**) Volcano plots illustrating the differentially expressed genes within the “defense response to virus” gene set (**D**), “negative regulation of viral process” gene set (**E**), “negative regulation of viral genome” gene set (**F**), and “regulation of viral entry into the host cell” gene set (**G**). (**H**) Venn diagram illustrating the expression of overlapping genes within each viral gene set (**D**–**G**) overexpressed in the SOX10 knockout populations.

**Table 1 cells-13-00073-t001:** Common differentially expressed genes from the A375 SOX10 knockout cells found within each viral gene set.

Process	Number of Genes	Gene List
Defense response to virus, Negative regulation of viral genome replication, Negative regulation of viral process, Regulation of viral entry into host cell	4	IFITM1, IFITM2, IFITM3, TRIM6
Defense response to virus, Negative regulation of viral process, Regulation of viral entry into host cell	5	TRIM31, MID2, TRIM21, TRIM8, TRIM56
Defense response to virus, Negative regulation of viral genome replication, Negative regulation of viral process	14	MAVS, APOBEC3F, ISG20, OASL, APOBEC3G, APOBEC3H, IFIH1, APOBEC3D, OAS2, ISG15, SHFL, RNASEL, APOBEC3C, BST2
Defense response to virus, Negative regulation of viral process	2	STAT1, TRIM32
Defense response to virus, Regulation of viral entry into host cell	1	TRIM38
Negative regulation of viral genome replication, Negative regulation of viral process	3	LTF, SLPI, HMGA2
Negative regulation of viral process, Regulation of viral entry into host cell	1	GSN
Negative regulation of viral process	3	SP100, ZFP36, SRPX2
Regulation of viral entry into host cell	7	TRIM34, NECTIN2, TMPRSS2, SMPD1, FURIN, BSG, LGALS1
Defense response to virus	57	IFI44L, DDX60, F2RL1, PTPN22, PMAIP1, CD86, IFIT1, TBK1, CREB3, DTX3L, ZDHHC12, VAMP8, TNFAIP3, CASP1, ATG7, NMB, TICAM2, KCNJ8, STAT2, UNC93B1, HTRA1, CARD9, IL6, CPTP, MOV10, TICAM1, ZC3H12A, TRIM7, MARCHF2, EXOSC4, IFNE, RIOK3, LYST, NCR1, IFNLR1, STING1, IFNA1, AIM2, PML, NLRX1, IL1B, ITGAX, GPAM, IRF1, IRF3, RNF185, UNC13D, LAMP2, SERTAD3, PARP9, IFIT3, USP20, IL10RB, NLRP3, CXADR, IRF7, ABCC9

## Data Availability

Data are contained within the article and Appendix A.

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
