# Peer review of "Sox10-Deficient Drug-Resistant Melanoma Cells Are Refractory to Oncolytic RNA Viruses"

_cells, 2023, doi:10.3390/cells13010073_

Round 1

Reviewer 1 Report

Comments and Suggestions for Authors

1. In the abstract, ISGs stand for interferon-induced genes, which the author did not include.

2. Regarding RNA-sequencing analysis, author mentioned that “Of the top 14 enriched biological processes in the SOX10-deficient A375, two were associated with the negative regulation of viral infection (Fig. 4B). We further mined any significant biological processes that had any impact on viral susceptibility and propagation and identified 14 of them (Fig. 4C).” The methods used by the author to identify significant biological processes affecting viral susceptibility and propagation are unclear.

Comments on the Quality of English Language

None

Reviewer 2 Report

Comments and Suggestions for Authors

1.This research focused on Sox10-deficient drug resistant melanoma cells are refractory to oncolytic RNA viruses , after check the pubmed, there were  not so many articles aboult this topic, so this manuscript was very prospective and significant.

2.This manuscript foucus on clinical problems of cancer, with strong clinical value and importantce,very interesting research,and also met the submission topic of this journal,the results was real and the conclusion was convincing, but some places can be more perfect.

3. In Material and Methods section, not found the Statistical Analysis part.

4. Targeted therapy resistant transition was very interesting and promising, how can use this service for clinic?

5.Figure 3B raw date revealed you have repeated 3 times, very good and rigorous research,if can do some Statistical analysis maybe much more better.

6.If not for oncolytic RNA viruses, Sox10-deficient can also have the same role? 

Comments on the Quality of English Language

Nearly ok.
